# Genetic Evolution and Variation of Human Adenovirus Serotype 31 Epidemic Strains in Beijing, China, during 2010–2022

**DOI:** 10.3390/v15061240

**Published:** 2023-05-25

**Authors:** Liying Liu, Yuan Qian, Zhenzhi Han, Liping Jia, Huijin Dong, Linqing Zhao, Runan Zhu

**Affiliations:** Laboratory of Virology, Beijing Key Laboratory of Etiology of Viral Diseases in Children, Capital Institute of Pediatrics, 2 Yabao Road, Beijing 100020, China; liuliying1201@163.com (L.L.);

**Keywords:** human adenovirus 31, diarrhea, molecular revolution, genetic diversity, children

## Abstract

Human adenovirus serotype 31 (HAdV-31) is closely associated with gastroenteritis in children and can cause fatal systemic disseminated diseases in immunocompromised patients. The lack of genomic data for HAdV-31, especially in China, will greatly limit research on its prevention and control. Sequencing and bioinformatics analyses were performed for HAdV-31 strains from diarrheal children in Beijing, China, during 2010–2022. Three capsid protein genes (hexon, penton, and fiber) were obtained in 37 cases, including one in which the whole genome was sequenced. HAdV-31 strains clustered into three distinct clades (I–III) in a phylogenetic tree constructed based on concatenated genes and the whole genome; the endemic strains only gathered into clade II, and most of the reference strains clustered into clade I. Compared with penton and hexon, fiber had a faster evolutionary rate (1.32 × 10^−4^ substitutions/site/year), an earlier divergence time (1697), lower homology (98.32–100% at the amino acid level), and greater genetic variation (0.0032). Four out of the six predicted positive selection pressure codons were also in the knob of fiber. These results reveal the molecular evolution characteristics and variations of HAdV-31 in Beijing, and fiber may be one of the main evolution driving forces.

## 1. Introduction

Human adenoviruses (HAdVs) are double-stranded DNA viruses without an envelope and with icosahedral capsids. Currently, there are over 111 identified types, which are classified into seven subgenera (A–G; http://hadvwg.gmu.edu/ (accessed on 28 February 2023)). After HAdV-52, adenoviruses are assigned by genomic and bioinformatic analyses instead of the traditional serum neutralization and hemagglutination inhibition test [1].

HAdV particles mainly consist of 240 hexons, 12 pentons, and 12 fibers. Hexons are considered type- and species-specific antigens, which contain most antigen epitopes in the hypervariable regions (HVR1–7) [2] and are highly sensitive to the pressure of immune selection [3]. Penton bases and fibers play a key role in viral adhesion and entry. The interaction between penton bases and integrins of a host cell promotes endocytosis, which is mediated by their RGD (Arg–Gly–Asp) motif [4]. Cell receptors mainly bind to fibers that have type-specific antigenic epitopes [5,6]. The complete genomic sequence of HAdV-31 was published in 2009. Some unique sequence features were conserved in all analyzed clinical isolates, such as the immunoglobulin-like domain of the E3 CR1 beta protein and a RGD motif located in protein IX [7].

A variety of clinical syndromes can be caused by HAdVs, mainly based on their tissue tropism and virulence [8]. Acute gastroenteritis is one of the most common manifestations in children. Aside from HAdV-40 and HAdV-41 in subgenus F, HAdV-31 is the most common adenovirus causing diarrhea in children, with an infection rate of 1.8% [9,10]. Serological studies have shown a high prevalence of HAdV-31 among persons of all age groups, especially in young children [7,11]. HAdV-31 belongs to subgenus A, which also includes HAdV-12, HAdV-18, and HAdV-61, recombined by serotypes 12 and 31 in the hexon gene [12]. Adenoviral diarrhea is usually not severe, but fatal systemic disseminated diseases can be caused by enteric infections with HAdV-31 in immunocompromised patients, especially in children undergoing hematopoietic stem cell transplantation (HSCT) [13,14]. There is no available adenovirus vaccine and no approved anti-adenoviral drugs. Recently, a study from Germany showed that HAdV-31 can be divided into six lineages with slow evolution [15]. Additionally, higher diversity gene regions in non-structural and minor capsid proteins can be used for separating lineages. However, so far, the molecular genetic characteristics and evolutionary history of endemic HAdV-31 strains in China remain unclear. The lack of genomic data will limit research on HAdV-31 prevention, control, and vaccines.

In this study, HAdV-31-positive stool specimens were selected from diarrheal children in Beijing, China, during 2010–2022, and three capsid protein genes (hexon, penton, and fiber) were amplified and sequenced in 37 cases, including one case in which the whole genome was sequenced. These sequences were analyzed for phylogenetic, evolutionary, and selection pressure analyses and compared with the sequences of HAdV-31 reference strains deposited in GenBank. The epidemiological characteristics and gene evolutionary characteristics of HAdV-31 endemic strains were revealed.

## 2. Materials and Methods

### 2.1. Sequencing of the Three Capsid Proteins of the HAdV-31 Endemic Strains and Complete Genome of the HAdV-31 Isolate

Stool samples were collected from children with acute diarrhea during 2010–2022 in Beijing, China. Nucleic acids of fecal specimens were extracted by using DNAzol (MRC, Inc., Cincinnati, OH, USA) and the universal primers were used in a polymerase chain reaction for detecting all types of HAdV [16,17]. Then, the PCR products were sequenced and analyzed for typing using the NCBI website. Three capsid protein genes were directly amplified from positive HAdV-31. Table 1 lists the primer pairs and amplification procedures designed for three genes: the penton base (P) gene, the hexon (H) gene, and the fiber (F) gene. The amplification products were sequenced by the Sino GenoMax Co., Ltd. (Beijing, China). Sequence splicing and assembling were carried out with DNAMAN 9.0.

An epidemic strain of HAdV-31 in Beijing was isolated in HEK-293 cells (Cell Resource Center, Peking Union Medical College) and passaged three times. DNA was extracted from 400 μL cell culture supernatant using the QIAamp DNA Blood Midi Kit (Qiagen, Hilden, Germany), and whole-genome sequencing (WGS) was conducted on an Illumina MiSeq platform (Illumina, San Diego, CA, USA) based on the standard protocols by the Magigene Company (Guangzhou, China). Briefly, extracted DNA was sheared by acoustic sonication (Covaris e220, CovarisInc, Woburn, MA, USA), DNA fragments underwent end-repair, A’-tailing and adaptor ligation, while fragments of approximately 350 bp were collected by beads after electrophoresis. According to the Covaries Illumina TruSeq DNA preparation protocol, libraries were pooled and subjected to 150 bp paired-end sequencing using the Novaseq 6000 platform (Illumina, San Diego, CA, USA). Sequence data were trimmed using the Fastp program [18], and human reads were removed using Bowtie2 [19]. Non-human reads were assembled using Trinity (version 2.8.4) and were taxonomically annotated using Kraken2 (version 2.0.8-beta) [20,21]. Finally, genomes were polished by the BLASTn package, and genome termini were manually checked and corrected using a map of the reads against the HAdV-A31 reference GenBank sequence (AM749299.1).

The sequences were confirmed by a BLAST comparison (http://www.ncbi.nlm.nih.gov/blast/Blast.cgi (accessed on 12 January 2021–21 December 2022)) and deposited into the GenBank database under different accession numbers: P gene (OQ657057–OQ657093), H gene (OQ657094–OQ657130), F gene (OQ657131–OQ657167), and one HAdV-31 whole genome RV388 (OQ744647). Meanwhile, eighteen reference strains with the whole genome sequence were selected from the GenBank database (listed in Table 2). The three capsid protein genes of the HAdV-61 Beijing strains were also used as reference sequences: P gene (OQ710460–OQ710463), H gene (OQ710456–OQ710459), and F gene (OQ710452–OQ710455).

### 2.2. Phylogenetic and Evolutionary Analyses of HAdV-31 Genes

Multiple sequence alignment of genes was performed using the MAFFT online service (https://www.ebi.ac.uk/Tools/msa/mafft/ (accessed on 11 January 2023)) [22]. Using the Kimura two-parameter model, the homology and evolutionary divergence of these sequences were estimated in MEGA 7.0 [23], and the phylogenetic trees of each gene and their concatemers and WGS were constructed using the maximum likelihood (ML) method with 1000 bootstrap replicates. Similarity plots (SimPlots) and recombination analyses were carried out by SimPlot 3.5.1 with default parameters [24].

The Markov chain Monte Carlo (MCMC) method was used to estimate the evolution rate (substitutions/site/year) and the time to the most recent common ancestor (TMRCA) in BEAST v 1.8.2 [25]. The nucleotide substitution models were generated in jModelstest v 2.1.7 [26], and the models with the highest Akaike Information Criterion (AIC) scores were selected: model HKY + I (P gene), model GTR + I (H gene), model GTR + I (F gene), and model GTR + I + G (tandem gene). The quality of the posterior distribution of each setting was determined according to the effective sample size (ESS) using Tracer v 1.7 [27] with 10% burning. SequenceMatrix v 1.7.8 [28] was used to concatenate the three capsid protein genes of the endemic strains and the reference strains following the order P + H + F.

### 2.3. Selection Pressure Analysis of the Three Capsid Proteins of HAdV-31

The selection pressures were evaluated by the ratio of the nonsynonymous (dN) and synonymous (dS) substitution rates at amino acid sites (ω = dN/dS): positive selection pressure (ω > 1), negative selection pressure (ω < 1) or neutral selection pressure (ω = 1). All analyses were performed on the Datamonkey web server (www.datamonkey.org (accessed on 16 January 2023)) [29]. Five algorithms were implemented to detect negative (purifying) and positive (diversifying) selection at the codon level, including a single-likelihood ancestor counting (SLAC) method, a fixed-effects likelihood (FEL) method, a mixed-effects model of evolution (MEME) method, a fast unconstrained Bayesian application (FUBAR) method, and a branch-site unrestricted statistical test for episodic diversification (BUSTED) method. Positive (dN > dS) selections were predicted using the *p*-value (*p* < 0.1) or the posterior probability (Post. Pr > 0.9) for at least two of the above methods. Negative (dN < dS) selections were predicted using the *p*-value (*p* < 0.05) or the posterior probability (Post. Pr > 0.95) for at least two of the above methods.

## 3. Results

### 3.1. General Information of HAdV-31-Positive Children in This Study

The three capsid protein genes were successfully amplified in a total of 37 fecal samples from HAdV-31-positive children from 2010 to 2022, which included 22 samples from boys (spheres in Figure 1) and 15 samples from girls (rhombuses in Figure 1). Twenty-five cases were outpatient children (marked in blue in Figure 1), and twelve cases were hospitalized children (marked in red in Figure 1), with a median age of 10.7 months and 19.4 months, respectively. Diarrhea (17 cases) and leukemia (7 cases) were the main diagnoses for outpatient and inpatient children, respectively. In 2016 and 2017, no HAdV-31-positive samples were detected; instead, one HAdV-61-positive sample was identified each year.

### 3.2. Phylogenetic Analysis of the Three Capsid Protein Genes of HAdV31

The ML tree of three capsid protein genes and concatenated genes from Beijing, as well as the reference strains, all showed that the HAdV-31 endemic strains gathered into a cluster with two HAdV-31 reference strains, OM372572 and MZ983592 (Appendix A and Figure 2A), and were separate from the other reference strains. The homology and divergence of the concatenated genes were 99.56–100% and 0.0018, respectively, among the endemic strains of HAdV-31; however, a homology of 97.95–100% and a divergence of 0.0051 were found between the Beijing strains and the reference HAdV-31 strains. The ML tree constructed based on the whole genome of one endemic isolate, RV388, and 18 reference strains suggested that the HAdV-31 strains were clearly divided into three clades (seen in Figure 2B). Clade I included the HAdV-31 prototype strain (AM749299) and most of the reference strains, clade II comprised one endemic strain (RV388) and two reference strains (OM372572 and MZ983592), and clade III was composed of the reference strains MW686774 and MZ983596. There was 98.71–100% homology and a total genetic distance of 0.0083 among the three clades. Among the three capsid proteins of HAdV-31, the highest amino acid variation was found in the fiber protein rather than in the penton and hexon proteins. The homology and differences of the three capsid protein genes among the endemic and reference strains of HAdV-31 are shown in Table 3.

### 3.3. Amino Acid Variation among the HAdV-31 Strains

The differences in the whole genome between the endemic strain RV388 and the representative reference strains from clades I to III of HAdV-31 are shown in Figure 3A–C. All sequences were aligned for similarity against the HAdV-31 prototype strain (GenBank acc. No. AM749299), and nucleotide similarities between RV388 and the reference strains from clades I to III were 98.87%, 99.92%, and 99.20%, respectively. RV388 almost overlapped with the other two reference strains in the same clade, and a few differences were observed between these three strains (Figure 3B,D). The variations between RV388 and the two reference strains in clade III were mainly located in the fiber protein, with up to 20 amino acid substitutions, and the other two amino acid substitutions were located in the hexon protein (Figure 3C,D). The above changes were conserved among the three strains of clade II. As shown in Figure 3A, the largest difference was found between RV388 and the reference strains of clade I. Five clade-specific amino acid substitutions were found in the three capsid proteins: G298N in penton, G415S in hexon, and N185D, S197N, and N296T in fiber. The last four of them were also specific amino acid substitutions of clade III.

### 3.4. The Selection Pressure Analysis of the Three Capsid Proteins of HAdV-31

In order to better understand the evolution dynamics of HAdV-31, selection pressure analyses of the three capsid proteins were performed based on the 37 endemic strains and 12 reference strains using five methods. The selection sites evaluated by at least two methods were considered to be significant, and Table 4 lists the specific positive and negative selection sites for the three proteins. All three capsid proteins contain positively selected sites. Codon 298 of penton under positive selection was the only different site between clade I and clade II + III (Figure 3D), which is located between the two motifs of penton binding to the cell receptor. Although the positive selection site 855 of hexon was the only different amino acid between the prototype strain (V) and the other HAdV-31 strains (L), both L and V are hydrophobic amino acids that support the protein structure. The four positive selection pressure sites of fiber are located in the knob region (380–556 aa), which can bind to the receptors of cells and has type-specific antigen epitopes. These sites showed the changes from hydrophilic amino acids (T and S) to hydrophobic amino acids (A) and from acidic amino acids (K and R) to basic amino acids (E and Q). Negative selection pressure was also found in all three proteins, of which hexon had the largest number of negative selective sites.

### 3.5. Evolutionary Genetic Analysis of the HAdV-31 Strains

Bayesian evolutionary analyses based on the tandem genes showed that HAdV-31 diverged from the common ancestors of HAdV-12 and HAdV-31 in 1875 (95% of the highest posterior density (HPD) in 1734–1957). Clade II was composed of HAdV-31 endemic strains that emerged in approximately 1968 (95% HPD, 1914–2002). The molecular evolutionary rate of the HAdV-31-concatenated genes was 7.86 × 10^−5^ substitutions/site/year (95% HPD, 1.99 × 10^−5^–1.4 × 10^−4^ substitutions/site/year) based on the data in these studies. As shown in Table 3 and Figure 4, hexon was the most conservative and stable gene with the slowest evolution rate and smallest amino acid variation among the three capsid genes of HAdV-31, whereas the fiber gene exhibited the fastest evolution rate, the earliest divergence time, and the most variation.

## 4. Discussion

HAdV-31 strains were originally isolated from the stools of seemingly healthy children [11]; however, many studies suggested that HAdV-31 is closely associated with gastroenteritis in children [9,30], and it was also the predominant pathogen detected in children with severe HAdV disease after HSCT [14,31,32]. Based on the surveillance of adenovirus in children with diarrhea, we found that HAdV-31 (1.8%) was the most common adenovirus aside from HAdV-41 (4.1%), especially in children younger than one year old or in children with leukemia (data not shown). In this study, more than half of the patients were younger than one year old, and more than half of the hospitalized patients suffered from leukemia, all in an immunocompromised state. These data indicate that HAdV-A31, as an etiological agent of a childhood disease, circulated continuously by infecting immunologically naive patients [15].

Our research and other studies showed that HAdV-31 has high genetic stability [14,15]. The nucleotide homologies of the three capsid proteins among epidemic strains in Beijing were higher than 99%. Due to the use of other types of adenoviruses in the same species as the out-group in the phylogenetic analysis, HAdV-31 strains clustered as three distinct clades in this study, whereas more lineages were presented when only HAdV-61 was used as the out-group in a study from Germany [15]. The HAdV-31 strains were clearly clustered into three clades in the ML tree based on the concatenated genes and whole genome. Clade I included the prototype strain AM749299 and reference strains from Germany, Britain, and Tunisia, which represent the strains of lineages 1–4 classified in reference [15]. The two HAdV-31 reference strains OM372572 (Germany) and MZ983592 (Germany) and all endemic strains in Beijing belonged to clade II, which was the same as lineage 6 reported in the above literature. In addition, clade III, identical to lineage 5, consisted of only two strains, MZ983596 (Germany) and MW686774 (UK). Compared with HAdV-41, another important pathogen of childhood diarrhea, the three capsid protein genes of HAdV-31 lacked significant intratypic genetic variability [33]. The length of each capsid protein was conservative, and no insertion or deletion of the amino acid was found. The variations in amino acids were mainly found in fiber, rather than in hexon, considered the main antigenic determinant protein. There were more than 18 clade-specific amino acid variations in the hexon of HAdV-41 [33], while the highly conserved hexon of HAdV-31 only had one such amino acid variation (G415S). This suggested that the changes in the epitopes related to immune escape may occur in the fiber of HAdV-31 rather than in the hexon, as reported for the other adenoviruses.

At present, there is no report on the evolution rate of HAdV-31. In this study, the evolution rates of the three capsid proteins and the tandem genes of HAdV-31 were estimated based on 12 reference strains selected from the GenBank database and 37 epidemic strains collected from 2010 to 2022 in Beijing. The average nucleotide evolution rate of HAdV-31 was 7.86 × 10^−5^ substitution/site/year, which is relatively slow. Among the three capsid proteins, fiber had the fastest evolution rate (1.32 × 10^−4^) with the earliest divergence, the lowest homology, and the greatest genetic variation. The evolution rate of the fiber of HAdV-31 was similar to that of HAdV-55 (1.238 × 10^−4^ substitutions/site/year) [34], higher than that of HAdV-41 (6.45 × 10^−5^) [33] and HAdV-4 (4.09 × 10^−5^) [35], but lower than that of HAdV-3 (1.085 × 10^−3^ substitutions/site/year) and HAdV-7 (0.132 × 10^−3^ substitutions/site/year) in subgenus B [36]. It was speculated that the fiber gene plays an important role in the evolution of HAdV-31.

The fiber of HAdV-31 had four positive selection pressure sites in this study: 556 (E/K), 441 (Q/R/P), 554 (T/A), and 431 (A/T/S), which were found in the knob region (380–556 aa), binding to the cell receptor [37]. The change from hydrophilic amino acids (T, S) to hydrophobic amino acids (A) may alter the structure of the knob, while the conversion of acidic amino acids (K, R) into basic amino acids (E, Q) may change the distribution of charges in the knob and change the affinity to cell receptors [4]. It is possible that the changes described above may affect the fiber binding to the receptor, which may further have an impact on its tissue affinity and promote the possibility of HAdV-31 generating multiple tissue hybrid orientations [4]. Codon 298 of penton under positive selection is located between the two motifs of penton binding to the cell receptor, and its mutation may affect the conformation of the motif bound to the receptor, thus affecting the process of the virus entering the cell [7]. In this study, variations in amino acids between different clades were found on the shaft and knob of the fiber protein (51–379aa) and the seventh highly variable region of the hexon protein (G415S), which may change their antigen epitopes to escape host immunity, thus promoting the evolution of HAdV-31.

Persistent HAdV-31 infections are progressively associated with morbidity in immunocompromised adults and in children undergoing HSCT [38]. Recent research also reported that HAdV-31 was related to obesity [39] and acute flaccid paralysis [40], the mechanism of which needs to be studied further. The existence of frequent asymptomatic HAdV infections makes epidemiological surveillance difficult; therefore, continued and active surveillance of HAdV-31 is necessary. In this study, the samples from children in Beijing did not represent the true situation in China. Further investigation of the epidemiology and evolution of HAdV-31 in China will provide more comprehensive data for preventing and controlling adenoviral diarrhea.

## Figures and Tables

**Figure 1 viruses-15-01240-f001:**
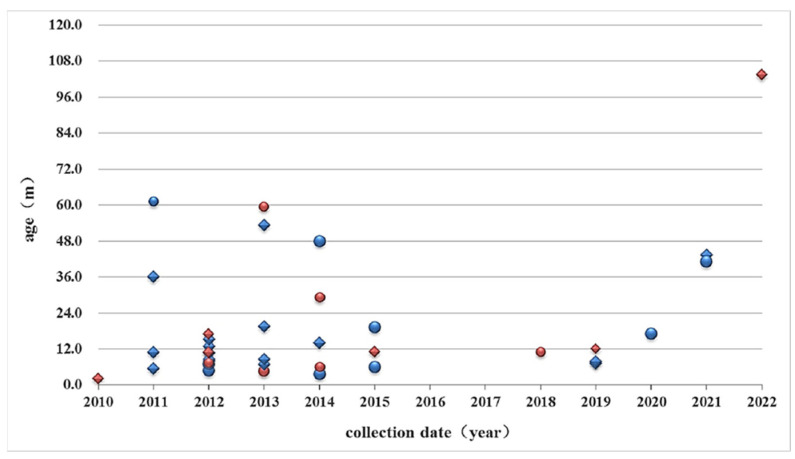
General information of HAdV-31-positive children. Boys and girls are represented by spheres and rhombuses, respectively. Outpatient children are marked in red and hospitalized children are marked in blue.

**Figure 2 viruses-15-01240-f002:**
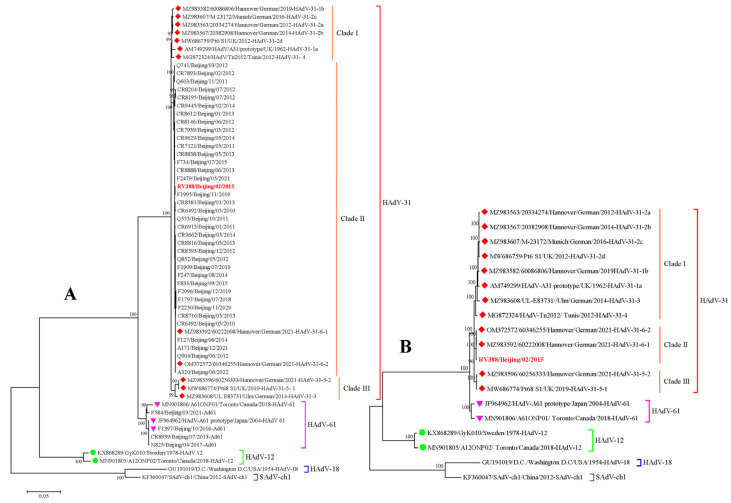
Phylogenetic tree of HAdV-A31 strains constructed using the ML method. The ML tree lists the concatenated genes (P + H + F) (**A**) and whole genome (**B**), with the HAdV-61, HAdV-12, HAdV-18, and SAdV-ch1 strains as reference strains for the out group. The reference strains of HAdV-31 are marked with diamonds, and one HAdV-31 endemic strain RV388 is marked in red. The reference strains of HAdV-61 and HAdV-12 are marked with a pink triangle and a green ball. The bootstrap values of ML are shown on the main clades.

**Figure 3 viruses-15-01240-f003:**
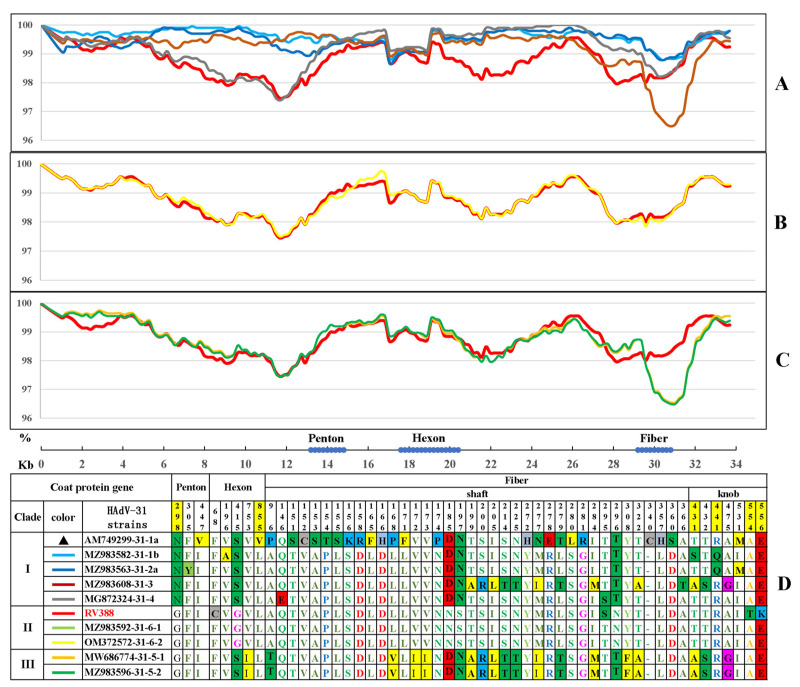
Similarity plot (SimPlot) of the whole genome and amino acid variation in three capsid proteins of HAdV-31. SimPlot of the nucleotide identity between RV388 and the representative reference strains of HAdV-31 for clade I (**A**), clade II (**B**), and clade III (**C**). Amino acid variations in the three capsid proteins among the HAdV-31 strains of the three clades are shown (**D**). The “▲” marks the HAdV-31 prototype strain (AM749299) used as query sequence in simplot analysis and different color lines repsent other representative strains of the three clades. The numbers (298, 855, 431, 441, 554, 556) highlighted in yellow in D represent amino acid sites under positive selection pressure.

**Figure 4 viruses-15-01240-f004:**
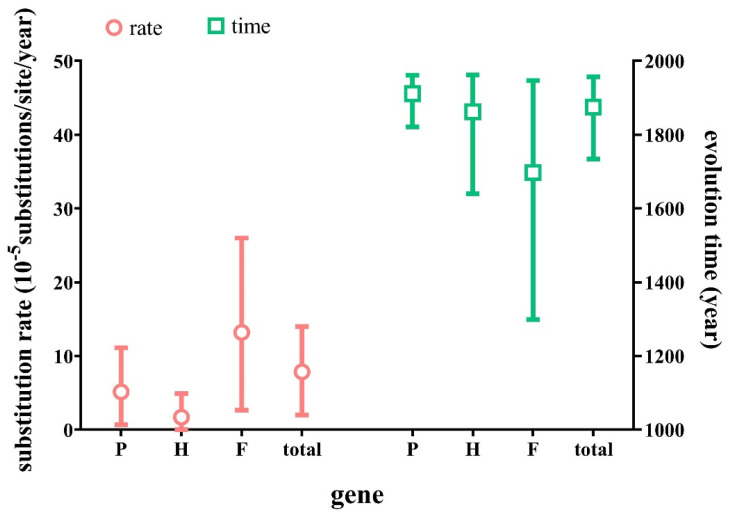
The estimated evolutionary rates and divergence time of the concatenated genes and three capsid protein genes of the HAdV-31 strains. The evolutionary rate of genes is displayed on the left, and the evolution time of genes is displayed on the right.

**Table 1 viruses-15-01240-t001:** Primers and amplification procedures used for the three genes of HAdV-31.

Genes	Primer Pairs		Position ^a^	Procedures				
				1 Cycle	30 Cycles			1 Cycle
				Preheat	Denaturation	Annealing	Extension	Incubation
Hexon	Forward	GATGGCCACTCCSTCGATG	17,574–17,592	94 °C 4 min	94 °C 40 s	58 °C 30 s	72 °C	72 °C 7 min
	Reverse	TCCTGYTCGCTTGAACCCAT	20,370–20,389				3 min	
Penton	Forward	CGACCAGYGTTCGTC	13,204–13,218			45 °C 30 s	72 °C	
	Reverse	CYGTGTTRTTACTTGGC	14,794–14,810				2 min	
Fiber 1	Forward	TTTCCYTCTTCCCAACT	29,038–29,054			45 °C 30 s	72 °C	
	Reverse	GGAGTTGTCCACARYGT	30,288–30,304				1 min 30 s	
Fiber 2	Forward	TGGAACTTAACAACTGATAT	30,096–30,115			40 °C 30 s	72 °C	
	Reverse	TKTTTTTATTCTTGGG	30,817–30,832				1 min	

^a^: The positions of the primers in the prototype strain of HAdV-31 (Ad31/UK/1962, GenBank acc. No. AM749299).

**Table 2 viruses-15-01240-t002:** The details of HAdV-31and other types of adenovirus reference strains.

Strains	Serotype	GenBank acc.no	Year	Country/Site	Annotation ^a^
Ad31	HAdV-31	AM749299	1962	UK	HAdV-A31 prototype(1a)
60086806	HAdV-31	MZ983582	2019	German/Hannover	1b
20334274	HAdV-31	MZ983563	2012	German/Hannover	2a
20382908	HAdV-31	MZ983567	2014	German/Hannover	2b
M-23172	HAdV-31	MZ983607	2016	German/Munich	2c
Pt6_S1	HAdV-31	MW686759	2012	UK	2d
UL-E83731	HAdV-31	MZ983608	2014	German/Ulm	3
HAdV-Tn2012	HAdV-31	MG872324	2012	Tunis	4
Pt68_S1	HAdV-31	MW686774	2019	UK	5-1
60256333	HAdV-31	MZ983596	2021	German/Hannover	5-2
60222008	HAdV-31	MZ983592	2021	German/Hannover	6-1
60346255	HAdV-31	OM372572	2021	German/Hannover	6-2
JPN/2004/5082	HAdV-61	JF964962	2004	Japan	HAdV-A61 prototype
A61ONP01	HAdV-61	MN901806	2018	Canada/Toronto	
A12ONP02	HAdV-12	MN901805	2018	Canada/Toronto	
GyK010	HAdV-12	KX868289	1978	Sweden	HAdV-A12 prototype
D.C	HAdV-18	GU191019	1954	USA/Washington D.C	HAdV-A18 prototype
SAdv-ch1	SAdV-ch1	KF360047	2012	China	Chimpanzee adenovirus

^a^: The numbers 1–6 represent the lineage of the HAdV-31 reference strains in reference [15].

**Table 3 viruses-15-01240-t003:** Evolutionary divergences among the three capsid protein genes of HAdV-31.

		37 Endemic Strains of HAdV-31			37 Endemic Strains and 12 Reference Strains of HAdV-31		
		P	H	F	P	H	F
Homology	DNA	99.60–100%	99.64–100%	99.04–100%	98.75–100%	99.06–100%	94.54–100%
	AA	99.60–100%	99.56–100%	98.32–100%	99.19–100%	99.45–100%	92.96–100%
Average Evolutionary Divergence	DNA	0.0013	0.0011	0.0032	0.0040	0.0030	0.0095
	AA	0.0011	0.0012	0.0050	0.0020	0.0014	0.0118

**Table 4 viruses-15-01240-t004:** Positive and negative selection codons in the three capsid proteins of HAdV-31 strains.

Protein	Positive Selection Sites	Negative Selection Sites
Penton	298(G/N)	28(Q), 248(F), 51(I), 55(E)
Hexon	855(L/V)	820(L), 453(D), 613(F), 716(K), 431(E), 657(E), 424(K), 900(Q), 827(Q)
Fiber	556(E/K), 441(Q/R/P), 554(T/A), 431(A/T/S)	553(I), 34(F),77(K), 27(Y), 157(L),134(L)

Note: *p* value of <0.1. Posterior probability of ≥0.95. All selected sites refer to the HAdV-31 prototype strain (AM749299).

## Data Availability

The datasets used in this study are available from the corresponding author on reasonable request. The complete genomic sequences and three capsid gene sequences of HAdV-A31 generated in this study are available at GenBank according to the accession numbers listed in this paper.

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
