# Peer review of "Genetic Evolution and Variation of Human Adenovirus Serotype 31 Epidemic Strains in Beijing, China, during 2010–2022"

_viruses, 2023, doi:10.3390/v15061240_

Round 1
Reviewer 1 Report
The work titled: “Genetic Evolution and Variation of Human Adenovirus Serotype 31 2 epidemic strains in Beijing, China, during 2010–2022” show relevant information about HAdV-31 evolution. The authors provide sequences of structural genes on various isolates from Beijing and one complete genome. This information together with that of other investigations gives us a glimpse of the panorama of the evolution of this serotype and its adaptation to its host. This last detail is of great importance due to the threat it represents as a pathogen. In general, the work is easy to read, the figures are clear and helpful, although the resolution of some of them could be improved. In some parts of the text is not clear if the authors compare RV388 isolate with the clades or they only compare the amino acids are conserved in the clade II with the rest one. The evolution rates for fiber HAdV-31 and other serotypes showed in discussion section is very interesting in order to speculate that this gene has great value in the evolution of HAdV-31
Minor Comments
Lines 35-36: “Cell receptors mainly bind to fibers that have type-specific antigenic epitopes [5,6]” The reference 5 does not mention interaction between cell receptor and fibers. Could you change it?
Lines 42-45: “Serological studies have shown a high prevalence of HAdV-31 among persons of all age groups, especially in young children [7, 11]” The reference 11 does not show information about HAdV-31. Could you change it?
Lines 68-69: “Sequence splicing and assembling were carried out with DNAMAN 9.0 [17]” The reference 17 describes MEGA 7, not DNAMAN. Could you correct it?
Lines 116-118: “SequenceMatrix v 1.7.8 [27] was used to concatenate the three capsid protein genes of the endemic strains and the reference strains following the order P + H + L.” I think it should be P + H + F, right?
Lines 174-176: “RV388 almost overlapped with the other two reference strains in the same clade, and no significant amino acid differences were found among the three capsid proteins (Figure 3B and 3D)” What does it mean that there are no significant amino acid differences? There are three changes hexon 68F/C, knob 553A/T and 555E/K which imply a change of charges or polarity of the amino acids. Maybe it would be more cautious to say that few differences are observed between these three references.
Lines 176-178: “The variation between RV388 and the two reference strains in clade III was mainly located in the fiber protein, with up to 20 amino acid substitutions, and the other two amino acid substitutions were located in the hexon protein” Looking closely at the figure 3D, I can see 4 additional changes (hexon 68F/C, shaft 295S/T, knob 553A/T and 555E/K) between RV 388 and the clade III references that are not mentioned. Or is it that you only want to mention the changes that are conserved in the three references of clade II. If so, could you clarify it in the text.
In the figure 3D, why are amino acids 298, 855, 430, 440, 553, 555 highlighted in yellow? You could explain in the text of figure.
In the table 4, I think there is a mistake when you mention fiber amino acid 440 (Q/R/P). In the figure 3D any proline is observed in the alignment.
Lines 213-214: “As shown in Figure 3 and Figure 4, hexon was the most conservative and stable gene…”. I think it would be better cited table 3 instead of figure 3, because in the alignment more changes are observed in hexon than penton. So it could give the false expectation that the penton is the most conserved.
Lines 225-226: “… we found that HAdV-31 (1.8%) was the most common adenovirus aside from HAdV-41 (4.1%), especially in children younger than one year old or in children with leukemia.” You could add the references for this information.
Lines 246-248: “There were more than 18 clade-specific amino acid variations in the hexon of HAdV-41, while the highly conserved hexon of HAdV-31 only had one such amino acid variation” You could add a reference where the 18 clade-specific amino acid variations for HAdV-41 is mentioned. You could tell which is the only amino acid in HAdV-31.
Lines 259-260: “… but lower than that of HAdV-3 (1.085×10-3 substitutions/site/year) and HAdV-7 (1.107×10-3 substitutions/site/year) in subgenus B [36].” There is a mistake in the value for HAdV-7. This value is the evolution rate of the hexon of HAdV-7 not fiber.
Lines 263-264: “… which were found in the knob region (380–555 aa), binding to the cell receptor [37].” Why do you cited this reference where HAdV-41 information is presented? According to HAdV-31 prototype (uniprot reference D0Z5U5), the knob region goes from 375 up 556.
Lines 272-273: “In this study, variations in amino acids between different clades were found on the shaft of the fiber protein (51–379).” I would add: variations in amino acids between different clades were found on the shaft and knob of the fiber protein. Attention, according to HAdV-31 prototype the knob fiber (uniprot reference D0Z5U5) starts at amino acid 375.
The authors could improve the resolution of figure 1 and 3D, because some numbers are difficult to read
Reviewer 2 Report
In their present manuscript, Liu et al. present an analysis of genetic data from human adenovirus type 31 samples collected in China between 2010 and 2022. This is an interesting virus genome study, and I only have a few questions and suggestions for minor changes:
It might be indicated in the introduction that not all 113 HAdV types are officially recognized by the ICTV (line 26-28).
In line 49-50, the authors cite a publication that showed slow evolution of Ad31, yet in the discussion in line 251 they state that there is no report on the evolution rate of HAdV-31. Please clarify.
It is not clear how the Ad31 infection was diagnosed in the first place, details should be provided. The referenced paper is in Chinese, therefore a brief description of the technique should be given.
All figures need legends, e.g. figure 1.
Lines 224-225: that is interesting data, but it is not shown nor referenced. I should at least say "data not shown".
The English Language is fine and might only need some minor editing.
